



**A global database of dissolved organic matter (DOM) measurements in coastal**
**waters (CoastDOM v1)**
Christian Lønborg[1*], Cátia Carreira[2], Gwenaël Abril[3], Susana Agustí[4], Valentina Amaral[5],
Agneta Andersson[6], Javier Arístegui[7], Punyasloke Bhadury[8], Mariana B. Bif[9], Alberto V.
Borges[10], Steven Bouillon[11], Maria Ll. Calleja[4,12], Luiz C. Cotovicz Jr[13,49], Stefano Cozzi[14],
Maryló Doval[15], Carlos M. Duarte[4], Bradley Eyre[16], Cédric G. Fichot[17], E. Elena García-
Martín[18], Alexandra Garzon-Garcia[19], Michele Giani[20,21], Rafael Gonçalves-Araujo[22],
Renee Gruber[23], Dennis A. Hansell[24], Fuminori Hashihama[25], Ding He[26], Johnna M.
Holding[27], William R. Hunter[28], J. Severino P. Ibánhez[29], Valeria Ibello[30], Shan Jiang[31],
Guebuem Kim[32], Katja Klun[33], Piotr Kowalczuk[34], Atsushi Kubo[35], Choon Weng Lee[36],
Cláudia B. Lopes[37], Federica Maggioni[38], Paolo Magni[39], Celia Marrase[40], Patrick
Martin[41], S. Leigh McCallister[42], Roisin McCallum[43], Patricia M Medeiros[44], Xosé Anxelu
G. Morán[4,45], Frank Muller-Karger[46], Allison Myers-Pigg[47], Marit Norli[48], Joanne M.
Oakes[15], Helena Osterholz[49], Hyekyung Park[32], Maria Lund Paulsen[50], Judith A.
Rosentreter[16,51], Digna Rueda-Roa[46], Chiara Santinelli[52], Yuan Shen[53], Eva Teira[54],
Tinkara Tinta[33], Guenther Uher[55], Masahide Wakita[56], Nicholas Ward[47], Kenta
Watanabe[57], Yu Xin[58], Youhei Yamashita[59], Liyang Yang[60], Jacob Yeo[16], Huamao
Yuan[61], Qiang Zheng[53,61], Xosé Antón Álvarez-Salgado[29]
[1]Section for Marine Diversity and Experimental Ecology, Department of Ecoscience,
Aarhus University, 4000 Roskilde, Denmark.
[2]Department of Environmental Science, Aarhus University, 4000 Roskilde, Denmark.
[3]Laboratoire de Biologie des Organismes et Ecosystèmes Aquatiques (BOREA), CNRS,
Muséum National d'Histoire Naturelle, 61 rue Buffon, 75005, Paris, France.
[4]King Abdullah University of Science and Technology, Thuwal 23955-6900, Kingdom of
Saudi Arabia.



[5]Departamento Interdisciplinario de Sistemas Costero Marinos, Centro Universitario
Regional Este, Universidad de la República, Ruta 9 y 15, CP 27000, Rocha, Uruguay.
[6]Umeå Marine Sciences Centre, Umeå University, Sweden.
[7]Instituto de Oceanografía y Cambio Global (IOCAG), Universidad de Las Palmas de
Gran Canaria, Las Palmas, Spain.
[8]Centre for Climate and Environmental Studies, Indian Institute of Science Education
and Research Kolkata, Mohanpur, Nadia, West Bengal, India.
[9]Monterey Bay Aquarium Research Institute, Moss Landing, California, United States.
[10]University of Liège, Chemical Oceanography Unit, Liège, Belgium.
[11]KU Leuven, Department of Earth and Environmental Sciences, Leuven, Belgium.
[12]Marine Ecology and Systematics (MarES), Department of Biology, Universitat de les
Illes Balears, 07122 Palma de Mallorca, Spain.
[13]Departamento de Geoquímica, Universidade Federal Fluminense, Outeiro São João
Batista s/n, 24020015 Niterói, RJ, Brazil.
[14]Consiglio Nazionale delle Ricerche, Istituto di Scienze Marine (CNR-ISMAR), Strada
Statale 14, km 163.5, 34149 Trieste, Italy.
[15]Instituto Tecnolóxico para o Control do Medio Mariño de Galicia, 36611 Vilagarcía de
Arousa, Spain.
[16]Centre for Coastal Biogeochemistry, Faculty of Science and Engineering, Southern
Cross University, Lismore 2480, NSW, Australia.
[17]Department of Earth and Environment, Boston University, Boston, MA, United States.
[18]National Oceanography Centre, European Way, Southampton, SO14 3ZH, United
Kingdom.
[19]Department of Environment and Science, PO Box 5078, Brisbane, Queensland 4001,
Australia.
[20]National Institute of Oceanography and Applied Geophysics (OGS), Trieste, Italy.



[21]Istituto Centrale per la Ricerca scientifica e tecnologia Applicata al MAre, Chioggia,
Italy
[22]National Institute of Aquatic Resources, Technical University of Denmark, Kgs.
Lyngby, Denmark.
[23]Australian Institute of Marine Science, PMB 3, Townsville QLD 4810, Australia
[24]Department of Ocean Sciences, Rosenstiel School of Marine and Atmospheric
Science, University of Miami, Miami, FL, United States.
[25]Tokyo University of Marine Science and Technology, Japan.
[26]Department of Ocean Science and Center for Ocean Research in Hong Kong and
Macau, The Hong Kong University of Science and Technology, Clear Water Bay, Hong
Kong, China.
[27]Department of Ecoscience, Aarhus University, 8000 Aarhus, Denmark.
[28]Fisheries and Aquatic Ecosystems Branch, Agri-Food and Biosciences Institute,
Belfast, Northern Ireland, United Kingdom.
[29]CSIC, Instituto de Investigacións Mariñas, Eduardo Cabello 6, 36208 Vigo, Spain.
[30]Institute of Marine Sciences, Middle East Technical University, 33731 Erdemli-Mersin,
Turkey.
[31]State Key Laboratory of Estuarine and Coastal Research, East China Normal
University, 200241, Shanghai, China.
[32]School of Earth and Environmental Sciences, Seoul National University, Seoul 08826,
Korea.
[33]Marine Biology Station, National Institute of Biology, Fornače 41, 6330 Piran,
Slovenia.
[34]Remote Sensing Laboratory, Institute of Oceanology, Polish Academy of Sciences,
Sopot, Poland.



[35]Department of Geosciences, Shizuoka University, 836 Ohya, Suruga-ku, Shizuoka,
422-8529, Japan.
[36]Laboratory of Microbial Ecology, Institute of Biological Sciences, Institute of Ocean
and Earth Sciences, Universiti Malaya, 50603 Kuala Lumpur, Malaysia.
[37]CICECO – Aveiro Institute of Materials, Department of Chemistry, University of Aveiro,
Campus de Santiago, 3810-193 Aveiro, Portugal.
[38]University of New Caledonia and Institute de recherche pour le development (IRD),
New Caledonia.
[39]Consiglio Nazionale delle Ricerche, Istituto per lo studio degli impatti Antropici e
Sostenibilità in ambiente marino (CNR-IAS), Loc. Sa Mardini, Torregrande, 09170,
Oristano, Italy.
[40]Institut de Ciències del Mar (CSIC), Passeig Marítim de la Barceloneta, 37
08003 Barcelona, Spain.
[41]Asian School of the Environment, Nanyang Technological University, 639798,
Singapore.
[42]Virginia Commonwealth University, Department of Biology, Richmond, Virginia, United
States.
[43]Centre for Marine Ecosystems Research, School of Science, Edith Cowan University,
Joondalup Drive, Joondalup 6027 WA, Australia.
[44]Department of Marine Sciences, University of Georgia, Athens, GA30602, United
States.
[45]Centro Oceanográfico de Gijón/Xixón, Instituto Español de Oceanografía, Gijón/Xixón,
Spain.
[46]College of Marine Science, University of South Florida, Saint Petersburg, Florida,
United States.



[47]Pacific Northwest National Laboratory, Marine and Coastal Research Laboratory,
Sequim, Washington, United States.
[48]Norwegian Institute for Water Research, Oslo, Norway.
[49]Leibniz Institute for Baltic Sea Research Warnemuende, D- 18119 Rostock-
Warnemuende, Germany.
[50]Marine microbiology, University of Bergen, Norway.
[51]Yale School of the Environment, Yale University, New Haven, Connecticut, United
States.
[52]Biophysics Institute, CNR, Pisa, Italy.
[53]State Key Laboratory of Marine Environmental Science (MEL) & College of Ocean
and Earth Sciences, Xiamen University, China.
[54]Departamento de Ecología y Biología Animal, Universidade de Vigo, Centro de
Investigacion Mariña da Universidade de Vigo(CIM-UVigo), Vigo, Spain.
[55]School of Natural and Environmental Science, Newcastle University, Newcastle upon
Tyne, United Kingdom.
[56]Mutsu Institute for Oceanography, Research Institute for Global Change, Japan
Agency for Marine-Earth Science and Technology, 690 Kitasekine, Sekine, Mutsu,
Aomori, Japan.
[57]Coastal and Estuarine Environment Research Group, Port and Airport Research
Institute, Yokosuka 239-0826, Japan.
[58]Key Laboratory of Marine Chemistry Theory and Technology, Ministry of Education,
Institute for Advanced Ocean Study, Ocean University of China, Qingdao, Shandong,
China.
[59]Faculty of Environmental Earth Science, Hokkaido University, Hokkaido 060-0810,
Japan.
[60]College of Environment and Safety Engineering, Fuzhou University, China.



[61]Fujian Key Laboratory of Marine Carbon Sequestration, Xiamen University, Xiamen
361102, China.

[*]Corresponding author:
E-mail: c.lonborg@ecos.au.dk or clonborg@gmail.com

**ORCID nr.:**
Christian Lønborg: 0000-0001-8380-0238
Cátia Carreira: 0000-0002-1520-9320
Gwenaël Abril: 0000-0002-4914-086X
Susana Agustí: 0000-0003-0536-7293
Valentina Amaral: 0000-0002-1088-1484
Agneta Andersson: 0000-0001-7819-9038
Javier Arístegui: 0000-0002-7526-7741
Punyasloke Bhadury: 0000-0001-8714-7475
Mariana Bernardi Bif: 0000-0002-2148-4556
Alberto V. Borges: 0000-0002-5434-2247
Steven Bouillon: 0000-0001-7669-2929
Maria Ll. Calleja: 0000-0002-5992-2013
Luiz C. Cotovicz Jr: 0000-0002-3914-8155
Stefano Cozzi: 0000-0003-0116-742X
Maryló Doval: 0000-0002-8565-8703
Carlos M. Duarte: 0000-0002-1213-1361
Bradley Eyre: 0000-0001-5502-0680
Cédric G. Fichot: 0000-0002-1099-5764
E. Elena García-Martín: 0000-0003-4807-3287
Alexandra Garzon-Garcia: 0000-0002-6804-8890
Michele Giani: 0000-0002-3306-7725
Rafael Gonçalves-Araujo: 0000-0001-8344-8326
Renee Gruber: 0000-0002-8788-6910
Dennis A. Hansell: 0000-0001-9275-3445
Fuminori Hashihama: 0000-0003-3835-7681
Ding He: 0000-0001-9620-6115
Johnna M. Holding: 0000-0002-7364-0055
William R. Hunter: 0000-0001-8801-7947
J. Severino P. Ibánhez: 0000-0001-6093-3054
Valeria Ibello: 0000-0002-1067-0425
Shan Jiang: 0000-0002-1121-6080
Guebuem Kim: 0000-0002-5119-0241
Katja Klun: 0000-0001-6111-1650
Piotr Kowalczuk: 0000-0001-6016-0610
Atsushi Kubo: 0000-0002-6457-5386
Choon Weng Lee: 0000-0001-9805-9980
Cláudia Lopes: 0000-0001-7378-8677
Federica Maggioni: 0000-0002-7109-4257



Paolo Magni: 0000-0001-5955-6829
Celia Marrase: 0000-0002-5097-4829
Patrick Martin: 0000-0001-8008-5558
S. Leigh McCallister: 0000-0002-9041
Roisin McCallum: 0000-0002-0358-2371
Patricia M Medeiros: 0000-0001-6818-2603
Xosé Anxelu G. Morán: 0000-002-9823-5339
Frank Muller-Karger: 0000-0003-3159-5011
Allison Myers-Pigg: 0000-0002-6905-6841
Marit Norli: 0000-0001-7472-1562
Joanne M. Oakes: 0000-0002-9287-2652
Helena Osterholz: 0000-0002-2858-9799
Hyekyung Park: 0000-0002-4743-5883
Maria Lund Paulsen: 0000-0002-1474-7258
Judith A. Rosentreter: 0000-0001-5787-5682
Digna Rueda-Roa: 000-0003-4621-009X
Chiara Santinelli: 0000-0002-8921-275X
Yuan Shen: 0000-0001-6618-4226
Eva Teira:  0000-0002-4333-0101
Tinkara Tinta: 0000-0001-6740-8973
Guenther Uher: 0000-0001-5105-4445
Masahide Wakita: 0000-0002-3333-0546
Nicholas Ward: 0000-0001-6174-5581
Kenta Watanabe: 0000-0002-0106-3623
Yu Xin: 0000-0002-5328-7717
Youhei Yamashita: 0000-0002-9415-8743
Liyang Yang: 0000-0001-8767-8698
Jacob Yeo: 0000-0003-2443-5378
Huamao Yuan: 0000-0003-2014-619X
Qiang Zheng: 0000-0002-6836-2310
Xosé Antón Álvarez-Salgado: 0000-0002-2387-9201





**Abstract**
The measurements of dissolved organic carbon (DOC), nitrogen (DON), and phosphorus
(DOP) are used to characterize the dissolved organic matter (DOM) pool and are
important components of biogeochemical cycling in the coastal ocean. Here, we present
the first edition of a global database (CoastDOM v1; available at
https://figshare.com/s/512289eb43c4f8e8eaef) compiling previously published and
unpublished measurements of DOC, DON, and DOP collected in coastal waters. These
data are complemented by hydrographic data such as temperature and salinity and, to
the extent possible, other biogeochemical variables (e.g., Chlorophyll-*a*, inorganic
nutrients) and the inorganic carbon system (e.g., dissolved inorganic carbon and total
alkalinity). Overall, CoastDOM v1 includes observations from all continents however,
most data were collected in the Northern Hemisphere, with a clear gap in coastal water
DOM measurements from the Southern Hemisphere. The data included were collected
from 1978 to 2022 and consist of 62339 data points for DOC, 20360 for DON and 13440
for DOP. The number of measurements decreases progressively in the sequence DOC
> DON > DOP, reflecting both differences in the maturity of the analytical methods and
the greater focus on carbon cycling by the aquatic science community. The global
database shows that the average DOC concentration in coastal waters (average
(standard deviation; SD): 182 (314) $\mu$mol C L$^{-1}$; median: 103 $\mu$mol C L$^{-1}$), is 13-fold greater
than the average coastal DON concentrations (average (SD): 13.6 (30.4) $\mu$mol N L$^{-1}$;
median: 8.0 $\mu$mol N L$^{-1}$), which was itself 39-fold greater than the average coastal DOP
concentrations (average (SD): 0.34 ± 1.11 $\mu$mol P L$^{-1}$; median: 0.18 $\mu$mol P L$^{-1}$). This
dataset will be useful to identify global spatial and temporal patterns in DOM and to
facilitate reuse of DOC, DON and DOP data in studies aimed at better characterising local
biogeochemical processes, closing nutrient budgets, estimating carbon, nitrogen and





phosphorous pools, as well as identifying a baseline for modelling future changes in
coastal waters.

**Keywords:** Dissolved organic matter, Dissolved organic carbon, Dissolved organic
nitrogen, Dissolved organic phosphorus, Coastal waters, Global database.





## 1. Introduction


Coastal waters are the most biogeochemical dynamic areas of the ocean, exhibiting
the highest standing stocks, process rates and transport fluxes of carbon (C), nitrogen
(N), and phosphorus (P) per unit area (Bauer et al., 2013; Mackenzie et al., 2011). In
these areas, organic matter plays a critical role in numerous biogeochemical processes,
serving as both a C, N and P reservoir and substrate (Carreira et al., 2021).
Organic matter found in the marine environment is commonly distinguished by its size;
material retained on a filter with a pore size typically between 0.2 and 0.7 µm is classified
as particulate organic matter (POM), whereas organic matter that passes through the filter
is referred to as dissolved organic matter (DOM). This partitioning is operational but has
implications for biogeochemical cycling: POM can be suspended in the water column or
sink to the sediments controlled by its size, shape and density (Laurenceau-Cornec et al.,
2015), whereas DOM is a solute that mostly remains in the water column. In most coastal
waters, the DOM concentrations are greater than POM, with the POM fraction being less
degraded and more bioavailable (Boudreau and Ruddick, 1991; Lønborg et al., 2018).
The DOM pool consists mainly of C (DOC), N (DON), and P (DOP) but it also includes
other elements such as oxygen, sulphur and trace elements (Lønborg et al., 2020). In
coastal waters, DOM originates from multiple sources. Internal, or autochthonous,
sources include planktonic organisms (Lønborg et al., 2009; Carlson and Hansell, 2015),
benthic microalgae, macrophytes, and sediment porewater (Burdige and Komada, 2014;
Wada et al., 2008 ). On the other hand, DOM from external, or allochthonous, sources,
has mainly terrestrial origins, including wetlands, river and surface runoff, groundwater
discharges, and atmospheric deposition (Iavorivska et al., 2016; Raymond and Spencer,
2015; Taniguchi et al., 2019; Santos et al., 2021). The main sinks for DOM from the water
column in coastal waters are: 1) bubble coagulation and abiotic flocculation (Kerner et al.,
2003) or sorption to particles (Chin et al., 1998); 2) sunlight mediated photodegradation



(Mopper et al., 2015); and 3) microbial degradation by mainly heterotrophic prokaryotes
(Lønborg and Álvarez-Salgado, 2012).
Given the importance of DOM as a source of nutrients and for coastal biogeochemical
cycling in general, numerous studies have measured the C, N and P content of the DOM
pool over the last few decades (e.g., García-Martín et al., 2021; Cauwet, 2002; Osterholz
et al., 2021). Most data, however, are often unavailable or stored in an inaccessible
manner, making it difficult to e.g., analyse global spatial and temporal patterns effectively.
A global open ocean DOM data compilation already exists, but it contains few coastal
samples (< 200m) (Hansell et al., 2021). Hence, there is a clear need for a comprehensive
global and integrated database of DOC, DON and DOP measurements for coastal waters.
To address this need, we have prepared the first edition of a coastal DOM database
(named CoastDOM v1), by compiling both previously reported as well as unpublished
data. These data have been obtained from authors of the original studies or extracted
directly from the original studies. In order to allow the DOM measurements to be
interpreted across larger scales, and to better understand their relationship with local
environmental conditions, we have included concurrently collected ancillary data (such
as physical and/or chemical seawater properties) whenever available. The objective of
this database is multifaceted. Firstly, we aimed to compile all available coastal DOM data
into a single repository. Secondly, our intention was to make these data easily accessible
to the research community and thirdly, we sought to achieve long-term consistency of the
measurements, to enable data intercomparison, and establish a robust baseline for
assessing, for example, the impacts of climate change and land use changes.

**2. Methods**
**2.1.   Data compilation**



The measurements included in CoastDOM v1 were obtained either directly from
authors of previously published studies, online databases, or scientific papers. An
extensive search of published reports, Ph.D. theses, and peer-reviewed literature was
performed to identify studies dealing with DOM in coastal waters. First, a formal search
was performed using Google Scholar in January 2022 using the search terms "dissolved
organic carbon", "dissolved organic nitrogen", and "dissolved organic phosphorus" in
connection with "marine" or "ocean", which yielded a total of 897 articles (after filtering
the query by searching content in the title and abstract and excluding non-coastal
articles). When data could not be obtained directly from the corresponding authors,
relevant data were extracted. Further searches for relevant datasets were conducted
using the reference lists of the identified scientific papers as well as databases and
repositories to capture as many datasets as possible. Additionally, research groups that
were invited to participate in this effort were also encouraged to submit unpublished data
to CoastDOM v1.

### 2.2.   Dissolved organic matter analysis
The DOC concentrations included in CoastDOM v1 were commonly measured using a
total organic carbon (TOC) high temperature catalytic oxidation (HTCO) analyser (81%
of samples). Some were measured by a combined wet chemical oxidation (WCO) step
and/or UV digestion, after which the carbon dioxide generated was quantified (19% of
samples). Similarly, concentrations of total dissolved nitrogen (TDN) were determined
using either a nitric oxide chemiluminescence detector connected in series with the HTCO
analyser used for DOC analyses (31% of the samples), or by employing a UV and/or
chemical oxidation step (69%). In the latter approach, both organic and inorganic N
compounds were oxidised to nitrate, which was subsequently quantified through a
colorimetric method to determine the concentration of inorganic N (Valderrama, 1981;



Álvarez-Salgado et al., 2023; Halewood et al., 2022; Foreman et al., 2019). The reported
DON concentrations were calculated as the difference between TDN and dissolved
inorganic nitrogen (DIN; sum of ammonium ($NH_4^+$) and nitrate/nitrite ($NO_3^-$ + $NO_2^-$); DON
= TDN - DIN) (Álvarez-Salgado et al., 2023). Analyses of total dissolved phosphorus
(TDP) were determined by UV (4%) or wet chemical oxidation (66%), or a combination of
these (30%), and subsequently were analysed for inorganic phosphorus by a colorimetric
method (Álvarez-Salgado et al., 2023). The DOP concentrations were calculated as the
difference between TDP and soluble reactive phosphorus (SRP: $HPO_4^{2-}$) (DOP = TDP -
SRP) (Álvarez-Salgado et al., 2023).

**3. Description of the dataset**
The data compiled in CoastDOM v1 were collected, analysed and processed by different
laboratories, however, all data included have undergone quality control measures, either
by using reference samples or internal quality assurance procedures. While many of the
included DOC and TDN data have been systematically compared against consensus
reference material (CRM) mainly provided by the University of Miami's CRM program
(Hansell, 2005), there is a limitation in CoastDOM v1 regarding the intercalibration across
different measurement systems used for both DOP and DON determination. While the
CRM could be used for DOC, DON and DOP measurements, this has not yet been
attempted for DOP and measurement uncertainties increase in the sequence DOC >
DON > DOP. Although some of the reported measurements have quantified the DOP
recovery based on commercially available DOP compounds such as Adenosine
triphosphate (ATP), it is not known if these were conducted systematically in all cases.
Therefore, we strongly recommend undertaking further intercalibrations across
laboratories for future measurements of TDP, as has been done for DOC and TDN
measurements (e.g., Sharp et al., 2002). Since additional quality control is not possible
in retrospect, we assessed the quality of CoastDOM v1 based on its internal consistency.

In CoastDOM v1, we defined "coastal water" as encompassing estuaries (salinity >

0.1) to the continental shelf break (water depth < 200 m). However, some locations, such
as deep fjords which are close to the coast cannot be classed as coastal due to
bathymetry (deeper than > 200 m). Therefore, we evaluated the inclusion of some
datasets on a case-by-case basis. For inclusion in the database, each DOM
measurement needed at a minimum to contain the following information (if reported in the
original publication or otherwise available):
- Country where samples were collected
- Latitude of measurement (in decimal units)
- Longitude of measurement (in decimal units)
- Year of sampling
- Month of sampling
- Sampling day (when available)
- Depth (m) at which the discrete sample were collected
- Temperature (°C) of the sample
- Salinity of the sample
- Dissolved organic carbon (DOC) concentration ($\mu$mol L$^{-1}$)
- Method used to measure DOC concentration
- DOC - QA flag: Quality flag for DOC measurement
- Dissolved organic nitrogen (DON) concentration ($\mu$mol L$^{-1}$)
- Total dissolved nitrogen (TDN) concentration ($\mu$mol L$^{-1}$)
- Method used to measure TDN concentration
- TDN - QA flag: Quality flag for TDN measurement
- Dissolved organic phosphorus (DOP) concentration ($\mu$mol L$^{-1}$)



- Total dissolved phosphorus (TDP) concentration ($\mu mol\ L^{-1}$)
- Method used to measure TDP concentration
- TDP - QA flag: Quality flag for TDP measurement
- Responsible person
- Originator institution
- Contact of data originator
It should be noted that in all entries, at least DOC, DON or DOP should have been
measured. In addition, we also included other relevant data, when available, in the
CoastDOM v1 dataset:
- Depth at the station where the sample was collected (Bottom depth, m).
- Total suspended solids (TSS) concentration ($mg\ L^{-1}$)
- Chlorophyll-*a* (Chl *a*) concentration ($\mu g\ L^{-1}$)
- Chl *a* - QA flag: Quality flag for chlorophyll-*a* measurement
- Sum of nitrate and nitrite ($NO_3^-+NO_2^-$) concentration ($\mu mol\ L^{-1}$)
- $NO_3^-+ NO_2^-$ - QA flag: Quality flag for $NO_3^-+ NO_2^-$ measurement
- Ammonium ($NH_4^+$) concentration ($\mu mol\ L^{-1}$)
- $NH_4^+$ - QA flag: Quality flag for $NH_4^+$ measurement
- Soluble reactive phosphorus ($HPO_4^{2-}$) concentration ($\mu mol\ L^{-1}$)
- $HPO_4^{2-}$ - QA flag: Quality flag for $HPO_4^{2-}$ measurement
- Particulate organic carbon (POC) concentration ($\mu mol\ L^{-1}$)
- Method used to measure POC concentration
- POC - QA flag: Quality flag for POC measurement
- Particulate nitrogen (PN) concentration ($\mu mol\ L^{-1}$)
- Method used to measure PN concentration
- PN - QA flag: Quality flag for PN measurement
- Particulate phosphorus (PP) concentration ($\mu mol\ L^{-1}$)



- Method used to measure PP concentration
- PP - QA flag: Quality flag for PP measurement
- Dissolved inorganic carbon (DIC) concentration (µmol kg$^{-1}$)
- DIC - QA flag: Quality flag for DIC measurement
- Total alkalinity (TA) concentration (µmol kg$^{-1}$)
- TA - QA flag: Quality flag for TA measurement

Quality control of large datasets is crucial to ensure their reliability and usefulness.
Thus, we have not included data that were deemed compromised, such as records that
had not gone through quality control by the data originators. We also accepted a certain
degree of measurement error since multiple groups have been involved in the collection,
analysis, and/or compilation of the information. Some of these errors were corrected (e.g.,
when a value was placed in a wrong column, or clearly inaccurate locations were
reallocated for consistency with the place of study), while others could not be rectified
(e.g., values showing clear signs of contamination) and were consequently excluded from
CoastDOM v1. It should also be noted that differences in analytical capabilities between
laboratories and individual measurement campaigns likely caused additional uncertainty.
Outliers, arising for example from contamination, were removed from the dataset. The
data were moreover screened for zero values (i.e., concentrations below the detection
limit or absence of data). In cases where concentrations were below the detection limit,
the zero values were replaced with half the value of the limit-of-detection. To ensure the
inclusion of only high-quality data, we only accepted entries with specific World Ocean
Circulation Experiment (WOCE) quality codes: "2- Acceptable measurement" and "6-
Mean of replicate measurements". In our quality control assessments, we carefully
avoided overly strict criteria, known as "data grooming", which could potentially overlook
genuine patterns and changes in the dataset that may be significant over longer temporal



and/or wider spatial scales. Coastal waters are known to exhibit a wide range of
environmental concentrations, influenced by factors such as seasonality and local
anthropogenic activities. Consequently, these data points may encompass a wide
concentration range.

**3.1 Summary of dissolved organic carbon (DOC) observations**
Measurements of DOC concentrations were conducted between 1978 to 2022, with a
total of 62339 individual data points (Table 1). The DOC concentrations ranged from 17
to 30327 µmol C L$^{-1}$ (average (Standard Deviation; SD): 182 (314) µmol C L$^{-1}$; median:
103 µmol C L$^{-1}$; Table 1). The majority (53%) of the concentrations fell within the range of
60 to 120 µmol C L$^{-1}$ (Fig. 1). A large number of DOC observations (17%) ranged between
300 and 600 µmol C L$^{-1}$, which were predominantly collected in eutrophic and river-
influenced coastal waters of the Northern Hemisphere, such as the Baltic Sea (Fig. 1). It
was observed that 75% of the DOC concentrations were higher than 77 µmol C L$^{-1}$, while
25% of the measurements surpassed 228 µmol C L$^{-1}$ (Table 1).
Coastal environments that experience minimal continental runoff, such as Palmer
Station in Antarctica, typically exhibit low DOC concentrations. On the other hand, coastal
waters heavily influenced by humic-rich terrigenous inputs, such as the Sarawak region
in Malaysia, tended to have high DOC concentrations. In addition some extreme high
DOC concentrations were measured in the Derwent River in Australia which is impacted
by paper mill effluents. There has been a large increase in the number of DOC
observations after 1992 (Fig. 2), and those measurements were from a wide range of
locations. However, these observations were not evenly distributed, with the Southern
Hemisphere being relatively under-sampled, especially in the African, South American
and Antarctic continents (Fig. 2, 3).





### 3.2. Summary of dissolved organic nitrogen (DON) observations


The DON measurements were collected between 1990 and 2021, with a total of 20357
data points (Table 1). Concentrations of DON ranged from < 0.1 to 2095.3 µmol N L$^{-1}$
(average (SD): 13.6 (30.4) µmol N L$^{-1}$; median: 8.0 µmol N L$^{-1}$; Table 1), with the most
common range (42%) for DON concentrations between 4 to 8 µmol N L$^{-1}$ (Fig. 1). Overall
75% of DON concentrations were above 5.5 µmol N L$^{-1}$, while 25% were above 15.8 µmol
N L$^{-1}$ (Table 1).
The lowest DON concentrations were recorded in Young Sound, Greenland, which
receives direct run-off from the Greenland Ice Sheet, whereas the highest concentrations
were detected during a flood event in the Richmond River Estuary, Australia. Since 1995,
there has been a large increase in the number of DON measurements conducted in
coastal waters globally (Fig. 2); however, the majority of those measurements have been
in the Northern Hemisphere, mostly in Europe and the United States (Fig. 2, 3).

### 3.3. Summary of dissolved organic phosphorus (DOP) observations


CoastDOM v1 includes a total of 13534 DOP measurements, collected between 1990
and 2021 (Table 1). Overall, DOP concentrations ranged from < 0.10 to 84.27 µmol P L$^{-1}$
(average (SD): 0.34 (1.11) µmol P L$^{-1}$; median: 0.18 µmol P L$^{-1}$; Table 2). The majority
(74%) of DOP concentrations were below 0.30 µmol P L$^{-1}$ (Fig. 1). Analysis of the DOP
dataset revealed that 75% of the concentrations were above 0.11 µmol P L$^{-1}$, while 25%
were above 0.30 µmol P L$^{-1}$ (Table 1).
The lowest DOP concentrations were measured off the Kimberley Coast in Australia,
while the highest concentrations were found in the Vasse-Wonnerup Estuary in the South
west region of Australia. Similarly to DOC and DON, most of the DOP measurements
have been conducted from the 1990s onwards, with a predominant focus in the Northern
Hemisphere, particularly in Europe and the United States (Fig. 2, 3).




### 3.4. Summary of dissolved organic matter (DOM) observations

In CoastDOM v1 the number of measurements decreases progressively in the
sequence DOC > DON > DOP, reflecting both differences in the maturity of the analytical
methods and the greater focus on carbon cycling by the aquatic science community. In
addition the average DOC concentration in coastal waters (182 (314) µmol C L$^{-1}$), was
13-fold greater than the average coastal DON concentrations 13.6 (30.4) µmol N L$^{-1}$),
which was itself 39-fold greater than the average coastal DOP concentrations (0.34 (1.11)
µmol P L$^{-1}$) (Table 1). Interestingly the coefficient of variation (C.V.) increased from DOC
(173%) to DON (224%) and DOP (326%), which is related to that the % contribution of
refractory organic material decreases in the same sequence (Table 1).

### 3.5. Potential use of the dataset

The use of the CoastDOM v1 dataset should be accompanied by the citation of this
paper and the inclusion of the correct doi-reference. CoastDOM v1 will be available in full
open access on the PANGEA homepage after acceptance of the manuscript, where it will
be available as a *.csv file. The dataset includes a brief description of the metadata and
methods employed, with emphasis on measurement techniques and data units. We
chose the terminology most familiar to the ocean science community. It is important to
note that all data included in CoastDOM v1, as well as this manuscript, are considered
public domain; as such, a subset of this global dataset may also be present in previous
data compilations (e.g., Hansell et al., 2021). The list of citations and links referenced in
CoastDOM v1 also provide users with information as to how these data has been
previously used in publications or databases.

### 3.6. Recommendations and conclusions

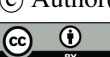



In CoastDOM v1, we have compiled available coastal DOM data in a single repository,
making it freely available to the research community. This compilation has established a
consistent global dataset, serving as a valuable information source to investigate a variety
of environmental questions and to explore spatial and temporal trends. We suggest a set
of recommendations for the future expansion of this global dataset. Firstly, our analysis
highlights a spatial bias, with a concentration of sampling efforts and/or data availability
predominantly concentrated in the Northern Hemisphere. The data gap in coastal DOM
measurements in the Southern Hemisphere needs to be addressed to provide a more
representative global understanding of the role of DOM in coastal water biogeochemistry.
Additionally, increased sampling efforts especially in the African and South American
continents are warranted due to the vulnerability of many coastal areas to climate change
and intensifying human activities, which will undoubtedly impact DOM biogeochemistry.
Further it is also worth noting that there is comparatively few data from coastal waters
affected by river discharge into the tropics, e.g., Amazon, Indian and Indonesian rivers
that together dominate freshwater inputs to the coastal ocean. Secondly, there is a need
for more comprehensive temporal and spatial datasets to capture the variability of DOM
levels in highly dynamic and productive coastal systems. Focused efforts should be made
to resolve these temporal and spatial changes. Thirdly, it is also important to collect and
report ancillary data, such as temperature, salinity, nutrient measurements, and
particulate components, to provide context and better understand the underlying
processes driving the observed DOM levels. Lastly, we strongly recommend that the
DOM research community conducts regular inter-calibration exercises to establish
standardised and interoperable methods, particularly for DON and DOP measurements.
This will ensure the comparability and reliability of data across different studies and
enhance our understanding of DON and DOP dynamics in coastal waters.



In light of ongoing global environmental changes, the mobilisation and open sharing of
existing data of important biogeochemical variables, such as the DOM pool, are crucial
for establishing baselines and determining global trends and changes in coastal waters.
The aim is to publish an updated version of the database periodically to determine global
trends of DOM levels in coastal waters, and we therefore encourage researchers to
submit new data to the corresponding author. The CoastDOM v1 dataset was developed
according to the FAIR principles regarding Findability, Accessibility, Interoperability and
Reusability of data. Thus, CoastDOM v1 will serve as a reliable open-source information
resource, enabling in-depth analyses and providing quality-controlled input data for large
scale ecosystem models.

**4. Data availability**
The       dataset     is     available     for     the     review     process     at     Figshare
https://figshare.com/s/512289eb43c4f8e8eaef). The dataset is furthermore submitted to
the PANGEA database and is currently waiting to be assigned a Doi number (Lønborg et
al., 2023). The file will be available as a *.csv merged file and will be available in full open
access in the PANGEA database after acceptance of the manuscript.

**Competing interests**
The authors declare no competing interests.
**Author Contribution**
C.L., C.C., and X.A.A-S started the initiative and finalised the data compilation. All co-
authors contributed data. C.L. wrote the manuscript with input from all co-authors.
**Acknowledgement**



Qi Chen is acknowledged for his very skilful help with Figure 3.
**Funding**
During the drafting of the manuscript C.L. received funding from the Independent
Research Fund Denmark Grant No. 1127-00033B. The monitoring data obtained from
Bermuda received funding from the Bermuda Government Department of Environment
and Natural Resources. A subset of the data obtained from UK coastal estuaries received
funding from the Natural Environment Research Council (Grant NE/N018087/1). Data
retrieved from the Palmer LTER data were collected with support from the Office of Polar
Programs, US National Science Foundation. Data obtained from the Great Barrier Reef
Marine Monitoring Program for Inshore Water Quality, which is a partnership between the
Great Barrier Reef Marine Park Authority, the Australian Institute of Marine Science,
James Cook University, and the Cape York Water Partnership. The contribution by P.K.
was supported by DiSeDOM project contract no. UMO-2019/33/B/ST10/01232 funded by
the NCN - National Science Centre, Poland. N.W. and A.M.-P. participated in this
synthesis effort with funding provided by the U.S. Department of Energy funded
COMPASS-FME project; the provided data was collected with funding from the PREMIS
Initiative, conducted under the Laboratory Directed Research and Development Program
at Pacific Northwest National Laboratory. The data obtained from the Levantine Sea (Med
Sea) received funding from the Scientific and Technological Research Council of Türkiye
(TÜBİTAK, 1001 program, Grant 115Y629).

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





**Figure legends**
**Figure 1.** Histograms showing the distribution of observations for **a**) dissolved organic
carbon (DOC), **b**) nitrogen (DON) and **c**) phosphorus (DOP), within defined
concentration ranges in the coastal ocean. Note that the concentration ranges are not
uniform in all cases due to the large difference in concentration levels.
**Figure 2**. **a**) Cumulative number of observations for dissolved organic carbon (DOC),
nitrogen (DON), and phosphorus (DOP). Number of observations shown as a function
of **b**) latitude, and **c**) longitude, grouped into bins of 10° latitude or longitude.
**Figure 3.** Global distribution of observations included in CoastDOM v1 for **a**) dissolved
organic carbon (DOC), **b**) nitrogen (DON), and **c**) phosphorus (DOP). The black dots
on the map represent the reported data that are included in the CoastDOM v1
database. Histograms show the distribution of observations for DOC, DON and DOP
within defined concentration ranges in the continents where measurements are
available. Maps were created using the GIS shape file obtained from Laurelle et al.

699 (2013)



**Table 1.** Descriptive statistics for the dissolved organic carbon (DOC), dissolved organic
nitrogen (DON), and dissolved organic phosphorus (DOP) measurements included in the
CoastDOM v1 dataset. The minimum (Min), maximum (Max), average values (Avg.) and
standard deviation (SD), coefficient of variation (CV %), median, 25th and 75th
percentiles (perc.) and number of samples (N) for each variable are shown.

| | DOC $\mu mol\ C\ L^{-1}$ | DON $\mu mol\ N\ L^{-1}$ | DOP $\mu mol\ P\ L^{-1}$ |
|---|---|---|---|
| Min | 17 | < 0.1 | < 0.01 |
| Max | 30327 | 2095.3 | 84.27 |
| Avg. (SD) | 182 (314) | 13.6 (30.4) | 0.34 (1.11) |
| Median | 103 | 8.0 | 0.18 |
| CV % | 173 | 224 | 326 |
| 25th perc. | 77 | 5.5 | 0.11 |
| 75th perc. | 228 | 15.8 | 0.30 |
| N | 62339 | 20357 | 13534 |






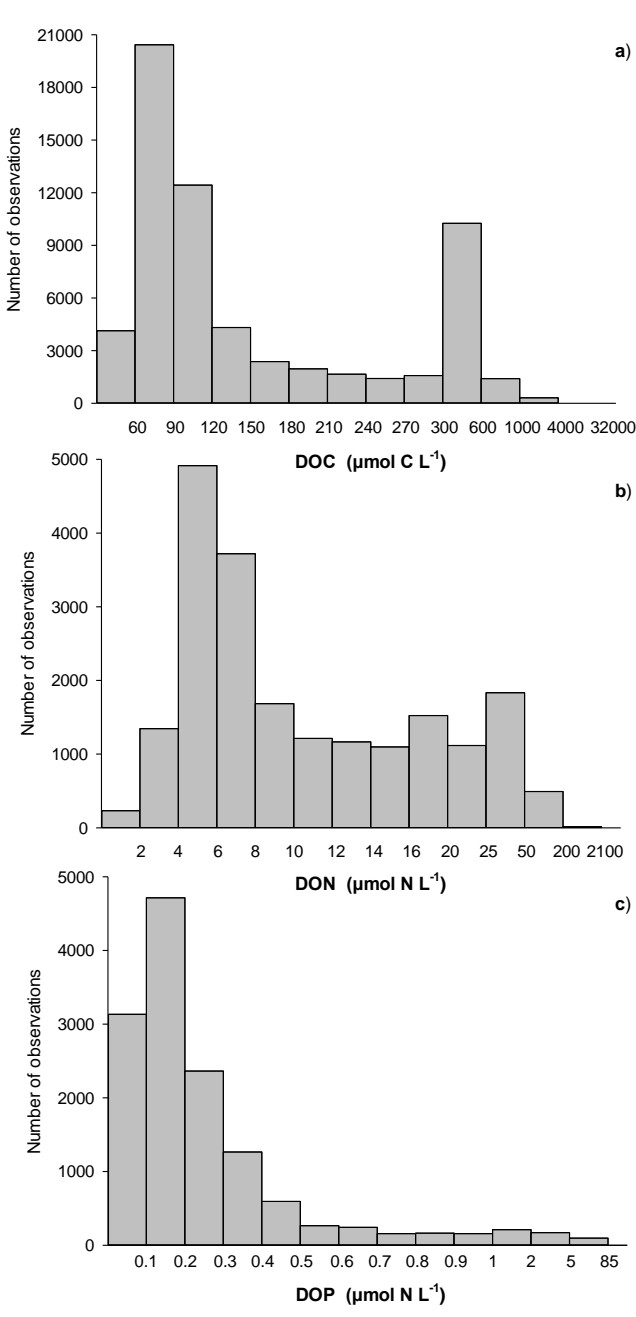


**Figure 1.**

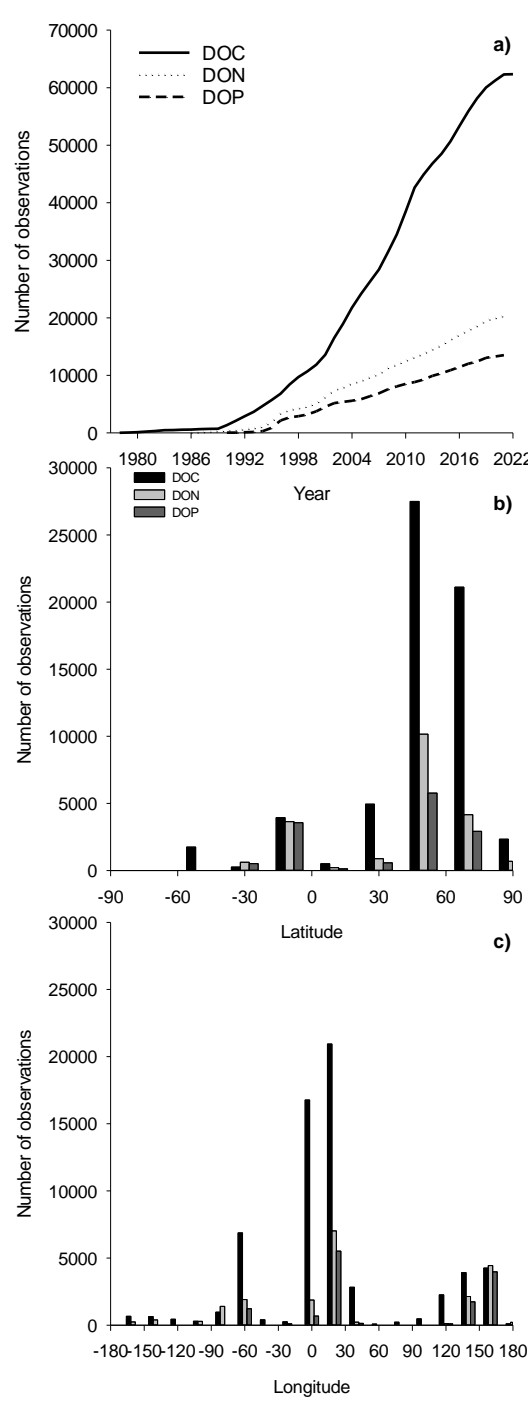


**Figure 2.**





**Figure 3.**