# Peer review of "A global database of dissolved organic matter (DOM) concentration"

_Earth System Science Data, 2023_

## Author Comment (AC1)

**Reviewer 2:**

*DOM represents a huge reservoir for biogenic elements and should be playing an important role in marine biogeochemical cycles. Given that most coastal waters are among the most dynamic and productive areas in the global ocean, a better understanding of DOM cycle in such environment is essential. However, there is clearly a huge gap between DOM data availability and its biogeochemical importance in coastal waters globally. To address the concern the authors here firstly present the most comprehensive database of DOM concentration in the global coastal waters to date, covering a wide range spatially and temporally as well. The aims of this work are clear and will be of interest to the wide readership of ESSD. It will also likely be a highly cited work considering the urgent call for such a comprehensive compilation of published+unpublished DOM concentration data. In general, the authors did a good job on data compilation. The manuscript is well written and well organized and will be a contribution to the aquatic science community.*

**Author reply:** We very much value the positive and constructive comments provided by the reviewer. Below we have made a detailed response to the comments point by point. Please note that the pages and lines from the reviewer correspond to those in the first submission. Pages, lines and figures numbers from our replies correspond to those in the revised version.

**Major comments**

*Title. The measurement of DOM can be multifaceted, i.e., concentration, chemical composition, bioavailability, age, isotopic ratios, etc. It's all about the most important parameter, i.e., concentration, in this effort of compilation.*

**Author reply:** You are right. We have changed the title to include the word 'concentration' as follows: A global database of dissolved organic matter (DOM) concentration measurements in coastal waters (CoastDOM v1)

*L279-281. I full agree with the authors that 'long-term consistency of the measurements, to enable data intercomparison, and establish a robust baseline for assessing,' is a fundamental issue. The authors have addressed the issue of comparability for data covering a wide range from multiple sources. The authors can provide more information (or give examples in SI) on this in the ms given its importance.*

**Author reply:** This is indeed an important point. Long term consistency can e.g., be achieved by these two ways: the CRM standards provided by the Hansell's laboratory, or by comparing the DOM concentrations obtained between laboratories in the same study area and time of year. This information was added as follows (L. 436-441):

"However, obtaining consistent long-term datasets is important to enable data intercomparison and establish a robust baseline. Such long-term consistency can be achieved by using the CRM standards provided by the Hansell laboratory for DOC and TDN. Another helpful approach is comparing the DOM concentrations obtained by different laboratories in the same study area and time of year."

*QC assessment part also deserves to be addressed in more detail. I would recommend a flow chart (step/criteria/priority) of data processing and quality control to be presented.*

*Are there any reliable time-series DOM concentration data included in this compilation that possibly provides a case to present or evaluate temporal variability/human perturbations, which is not generally lacking in its present form?*
**Author reply:** Regarding the flow chart, we have included a new figure (New Fig. 1) that shows the steps taken throughout the process from receiving the data to including it in the database.

There are some relatively long time series in the database, but the authors would prefer to keep this manuscript focused on the database instead of giving specific examples, as this would spread the message and would also require giving other specific examples (e.g., spatial distribution).

*I would recommend to try an addition of stoichiometry (ratios C/N/P) part. It may unveil the tight link for the elements and provide implications for understanding biogeochemical cycle in the highly dynamic and rapidly changing coastal waters.*
**Author reply:**  In Table 1 we now report the DOC:DON, DOC:DOP and DON:DOP ratios. We furthermore provide a brief overview of the stoichiometry in the manuscript (L.507-511):

"The average C: N: P stoichiometry for these paired DOM measurements was 1171 (± 4248): 100 (± 580): 1 (Table 1), which was very N- and P- depleted compared to the Redfield Ratio (Redfield et al., 1963). However, the large variations in C:N, C:P and N:P ratios reveals large variations in the composition of the DOM pool in coastal waters. "

**Minor comments**
*L407-408: 'In cases where concentrations were below the detection limit, the zero values were replaced with half the value of the limit-of-detection.' I am not sure if this is the best practice. What about leaving it blank?*
**Author reply:** We understand the comment and agree that this is not a perfect solution, but our aim here was to distinguish between no measure (blank), and a measure below detection limit (half of the detection limit). Therefore, we would like to keep these values in the database.

*Typo: The unit on x axis would read 'umol P L-1) in Fig.1c.*
**Author reply:** Corrected.

*The color is too light for DON line in Fig.2a.*
**Author reply:** We have changed the thickness of the lines in Fig.3a.

*Fig 3 is not clear enough, especially for histograms.*
**Author reply:** We have created a new figure which has a higher resolution.

---

## Author Comment (AC2)

**Reviewer 1:**
*The authors took tremendous efforts to collect published and unpublished DOC, DON and DOP concentration measurements from global coastal regions and provided a useful dataset to the marine biogeochemistry and DOM community. The dataset is well-organized, and the manuscript introduced this open-access dataset well. It may require some revisions that could help improve the paper.*

**Author reply:** We appreciate the positive and constructive comments provided by the reviewer and we hope to have addressed clearly and satisfactorily all the comments posed. Below we have made a detailed reply to the comments point by point. Please note that the page and line numbers from the reviewer correspond to the first submission. Pages, lines and figures numbers from our replies correspond to the revised manuscript.

**Suggestions:**
*Title: I suggest specifying what "dissolved organic matter (DOM) measurements" refer to in the title to make it clearer. I believe they are "Dissolved organic carbon (DOC), nitrogen (DON), and phosphorus (DOP) concentration measurements".*

**Author reply:** We have included 'concentration ' in the title as suggested.

*Lines 262-268: Hansell et al,. 2021 DOM data compilation only contains the TDN and DOC concentration measurements. It may be helpful if adding some discussions here about existing open ocean DOP concentration dataset such as an old DOP concentration database called "GOOD DOP" in a book chapter (Karl & Björkman, 2015) and a recently compiled open-access open ocean DOP concentration database (DOPv2021) (Liang et al., 2022).*
*Reference:*
*Karl, D. M. & Björkman, K. M. Chapter 5 - Dynamics of dissolved organic phosphorus. in Biogeochemistry of Marine Dissolved Organic Matter 2nd edn (eds. Hansell, D. A. & Carlson, C. A.) 233–334, https://doi.org/10.1016/B978-0-12-405940-5.00005-4 (Academic Press, 2015).*
*Liang, Z., McCabe, K., Fawcett, S.E. et al. A global ocean dissolved organic phosphorus concentration database (DOPv2021). Sci Data 9, 772 (2022). https://doi.org/10.1038/s41597-022-01873-7*

**Author reply:** You are right. We have included these additional references and reformulated the sentence, so it now reads (L.275-278):

"Global open ocean DOM data compilation for DOC total dissolved nitrogen (TDN) (Hansell et al., 2021) and DOP (Liang et al., 2022; Karl and Björkman, 2015) already exist and contains few coastal samples (< 200m) (Hansell et al., 2021), but there are no compilation specifically focused on coastal waters."

*Lines 290-293: Some articles might use key words such as "Total organic carbon", "Total dissolved nitrogen", "Total dissolved phosphorus", "Total organic nitrogen", and "Total organic phosphorus". I am not sure if only using the terms "dissolved organic carbon", "dissolved organic nitrogen", and "dissolved organic phosphorus" would miss some published data.*

**Author reply:** Yes, this is a good point. Initially we also tested for this but found that the same number of total papers were identified and we therefore chose to restrict our search terms.

*Lines 308-310: After oxidation of DON samples, the resulting total nitrate can also be measured by the nitric oxide chemiluminescence method (Knapp et al., 2005;) not just colorimetric method.*
*Reference:*
*Knapp, A. N., Sigman, D. M., & Lipschultz, F. (2005). N isotopic composition of dissolved organic nitrogen and nitrate at the Bermuda Atlantic Time-series Study site. Global Biogeochemical Cycles, 19(1).*
**Author reply:** You are right. We now mention this method and added the following sentence (L. 321-324):

"Another method used for DON determination is oxidizing the sample and measuring the resulting total nitrate by the nitric oxide chemiluminescence method (Knapp et al., 2005). However, none of the concentration measurements included in CoastDOM v1 applied this method."

*Lines 301-308: I believe it needs references here for the approaches for TOC and TDN concentration measurements. Below are some resources.*
*For TOC: Sharp, J. H., Benner, R., Bennett, L., Carlson, C. A., Dow, R., & Fitzwater, S. E. (1993). Re-evaluation of high temperature combustion and chemical oxidation measurements of dissolved organic carbon in seawater. Limnology and Oceanography, 38(8), 1774-1782.*
*For TDN: in the Chapter 4 of the book "Biogeochemistry of marine dissolved organic matter", it summarizes different oxidation approaches for DON and includes their references.*
**Author reply:** We thank the reviewer for the references which we have now included in the revised manuscript.

*Lines 314-317: For TDP analysis, besides UV and wet chemical oxidation, there is another oxidation approach "ash/hydrolysis". Details of the "ash/hydrolysis" approach can be found in Solórzano & Sharp 1980. However, perhaps no DOP data in this database employed the "ash/hydrolysis" approach?*
*Reference:*
*Solórzano, Lucia, and Jonathan H. Sharp. "Determination of total dissolved phosphorus and particulate phosphorus in natural waters." Limnology and Oceanography 25.4 (1980): 754-758.*
**Author reply:** Thanks for the suggestion. We have added this to the manuscript as follows (L.330-332):

"Another method also previously used for TDP analysis is the ash/hydrolysis method (Solorzano and Sharp, 1980), even though none of the data included in CoastDOM v1 used this method."

*Line 407: what is the detection limit here for DOC. DON and DOP concentration measurements?*
**Author reply:** Commonly reported detection limits are ~4 µmol C $L^{-1}$ for DOC, ~0.3 µmol N $L^{-1}$ for DON and are ~0.03 µmol P $L^{-1}$ for DOP. This information has been included in the revised manuscript (L.426-427).

*Lines 496-509: How many data points have all DOC, DON and DOP concentration measurements? The article says that observations of DON and DOP concentrations are much less than DOC concentrations. I would add a recommendation here to acknowledge this feature and encourage more paired measurements of DOC, DON and DOP concentrations.*

**Author reply:** Thanks for the suggestion. In CoastDOM v1 there were in total 7058 paired measurements of DOC, DON, and DOP. This information was added as follows (L. 505-507):

"It should be noted that CoastDOM v1 only contains 7058 paired measurements of DOC, DON, and DOP, and therefore only a subset of observations reported all three element pools. "

And l.543-545:

" Third, only a fraction of data entries report paired DOC, DON and DOP measurements, we encourage  that these be measured and reported  together in order to better determine changes in stoichiometry and composition. "

*Figures: Why not add a figure to show number of observations for dissolved organic carbon (DOC), nitrogen (DON), and phosphorus (DOP) concentrations in different months? I would also distinguish samples from the northern hemisphere and southern hemisphere. This figure will help readers check the potential bias due to sampling seasons.*

**Author reply:** We have now added a figure showing the number of observations for DOC, DON and DOP concentrations in different months (Fig. 3b).
Regarding differences in observations between the north and south hemisphere we have in the text added % of observations in either the Southern (10% of DOC observations; L. 461) or Northern Hemisphere (79% of DON (L. 477) and 70% of DOP (L. 492) observations).

**Minor comments:**
*Lines 205-206: I would add the word "concentrations" to clarify these measurements are DOC, DON and DOP concentrations. Similar for other texts in the article. For example, use "DON concentration observations" instead of "DON observations".*

**Author reply:** We have changed it here and further done our best to consistently use "concentration" throughout the manuscript.

*Line 241: I would use "organic material" here.*
**Author reply:** Changed accordingly.

*Lines 246-248: I find this sentence a bit difficult to follow. Why POM less degraded but more bioavailable?*

**Author reply:** The POM pool is generally more bioavailable than the DOM pool as generally this pool is less reworked and recalcitrant and contains a larger proportion of known biochemical classes (e.g., carbohydrates, proteins, lipids, nucleic acids). We reformulated the sentence to reflect this idea and provide more clarity (L.252-256):

"In most coastal waters, DOM concentrations are higher than POM, with POM having a larger proportion of known biochemical classes (e.g., carbohydrates, proteins) than the dissolved fraction, suggesting that generally, DOM is more reworked and recalcitrant (Boudreau and Ruddick, 1991; Lønborg et al., 2018; Benner and Amon, 2015)."

*Figure 1c: x- axis unit should be µmol P L$^{-1}$*
**Author reply:** Corrected.

**Reviewer 2:**
*DOM represents a huge reservoir for biogenic elements and should be playing an important role in marine biogeochemical cycles. Given that most coastal waters are among the most dynamic and productive areas in the global ocean, a better understanding of DOM cycle in such environment is essential. However, there is clearly a huge gap between DOM data availability and its biogeochemical importance in coastal waters globally. To address the concern the authors here firstly present the most comprehensive database of DOM concentration in the global coastal waters to date, covering a wide range spatially and temporally as well. The aims of this work are clear and will be of interest to the wide readership of ESSD. It will also likely be a highly cited work considering the urgent call for such a comprehensive compilation of published+unpublished DOM concentration data. In general, the authors did a good job on data compilation. The manuscript is well written and well organized and will be a contribution to the aquatic science community.*
**Author reply:** We very much value the positive and constructive comments provided by the reviewer. Below we have made a detailed response to the comments point by point. Please note that the pages and lines from the reviewer correspond to those in the first submission. Pages, lines and figures numbers from our replies correspond to those in the revised version.

**Major comments**
*Title. The measurement of DOM can be multifaceted, i.e., concentration, chemical composition, bioavailability, age, isotopic ratios, etc. It's all about the most important parameter, i.e., concentration, in this effort of compilation.*
**Author reply:** You are right. We have changed the title to include the word 'concentration' as follows: A global database of dissolved organic matter (DOM) concentration measurements in coastal waters (CoastDOM v1)

*L279-281. I full agree with the authors that 'long-term consistency of the measurements, to enable data intercomparison, and establish a robust baseline for assessing,' is a fundamental issue. The authors have addressed the issue of comparability for data covering a wide range from multiple sources. The authors can provide more information (or give examples in SI) on this in the ms given its importance.*
**Author reply:** This is indeed an important point. Long term consistency can e.g., be achieved by these two ways: the CRM standards provided by the Hansell's laboratory, or by comparing the DOM concentrations obtained between laboratories in the same study area and time of year. This information was added as follows (L. 436-441):

"However, obtaining consistent long-term datasets is important to enable data intercomparison and establish a robust baseline. Such long-term consistency can be achieved by using the CRM standards provided by the Hansell laboratory for DOC and TDN.

Another helpful approach is comparing the DOM concentrations obtained by different laboratories in the same study area and time of year."

*QC assessment part also deserves to be addressed in more detail. I would recommend a flow chart (step/criteria/priority) of data processing and quality control to be presented.*
*Are there any reliable time-series DOM concentration data included in this compilation that possibly provides a case to present or evaluate temporal variability/human perturbations, which is not generally lacking in its present form?*
**Author reply:** Regarding the flow chart, we have included a new figure (New Fig. 1) that shows the steps taken throughout the process from receiving the data to including it in the database.
There are some relatively long time series in the database, but the authors would prefer to keep this manuscript focused on the database instead of giving specific examples, as this would spread the message and would also require giving other specific examples (e.g., spatial distribution).

*I would recommend to try an addition of stoichiometry (ratios C/N/P) part. It may unveil the tight link for the elements and provide implications for understanding biogeochemical cycle in the highly dynamic and rapidly changing coastal waters.*
**Author reply:** In Table 1 we now report the DOC:DON, DOC:DOP and DON:DOP ratios. We furthermore provide a brief overview of the stoichiometry in the manuscript (L.507-511):

"The average C: N: P stoichiometry for these paired DOM measurements was 1171 (± 4248): 100 (± 580): 1 (Table 1), which was very N- and P- depleted compared to the Redfield Ratio (Redfield et al., 1963). However, the large variations in C:N, C:P and N:P ratios reveals large variations in the composition of the DOM pool in coastal waters. "

**Minor comments**
*L407-408: 'In cases where concentrations were below the detection limit, the zero values were replaced with half the value of the limit-of-detection.' I am not sure if this is the best practice. What about leaving it blank?*
**Author reply:** We understand the comment and agree that this is not a perfect solution, but our aim here was to distinguish between no measure (blank), and a measure below detection limit (half of the detection limit). Therefore, we would like to keep these values in the database.

*Typo: The unit on x axis would read 'umol P L-1) in Fig.1c.*
**Author reply:** Corrected.

*The color is too light for DON line in Fig.2a.*
**Author reply:** We have changed the thickness of the lines in Fig.3a.

*Fig 3 is not clear enough, especially for histograms.*
**Author reply:** We have created a new figure which has a higher resolution.

---

## Author Comment (AC3)

[revised manuscript text omitted]